# High-Frequency Deep Sclerotomy, A Minimal Invasive Ab Interno Glaucoma Procedure Combined with Cataract Surgery: Physical Properties and Clinical Outcome

**Bojan Pajic** [1,2,3,4,*] **, Zeljka Cvejic** [2] **, Kaweh Mansouri** [5,6] **, Mirko Resan** [4] **and Reto Allemann** [1]

1. Eye Clinic Orasis, Swiss Eye Research Foundation, 5734 Reinach AG, Switzerland; rallemann@gmail.com
2. Department of Physics, Faculty of sciences, University of Novi Sad, Trg Dositeja Obradovica 4, 21000 Novi Sad, Serbia; zeljka.cvejic@df.uns.ac.rs
3. Division of Ophthalmology, Department of Clinical Neurosciences, Geneva University Hospitals, 1205 Geneva, Switzerland
4. Faculty of Medicine of the Military Medical Academy, University of Defense, 11000 Belgrade, Serbia; resan.mirko@gmail.com
5. Glaucoma Center, Montchoisi Clinic, 1000 Lausanne, Switzerland; kwmansouri@gmail.com
6. Department of Ophthalmology, University of Colorado, Denver, CO 80014, USA
* Correspondence: bpajic@datacomm.ch; Tel.: +41-62-765-60-80

**Abstract:** Background: The efficiency and safety of primary open-angle glaucoma with high-frequency deep sclerotomy (HFDS) combined with cataract surgery has to be investigated. Methods: Right after cataract surgery, HFDS was performed ab interno in 205 consecutive patients with open angle glaucoma. HFDS was performed with a custom-made high-frequency disSection 19 G probe (abee tip 0.3 × 1 mm, Oertli Switzerland). The bipolar current with a frequency of 500 kHz is applied. The nasal sclera was penetrated repetitively six times through the trabecular meshwork and consecutively through Schlemm's canal. Every time, a pocket of 0.3 mm high and 0.6 mm width was created. Results: Mean preoperative intraocular pressure (IOP) was 24.5 ± 2.1 mmHg (range 21 to 48 mmHg). After 48 months, the follow up average IOP was 15.0 ± 1.7 mmHg (range 10 to 20 mmHg). Postoperative IOP has been significantly reduced compared to preoperative IOP for all studied cases ($p < 0.001$). After 48 months, the target IOP less than 21 mmHg reached in 84.9%. No serious complications were observed during the surgical procedure itself and in the postoperative period. Conclusions: HFDS is a minimally invasive procedure. It is a safe and efficacious surgical technique for lowering IOP combined with cataract surgery.

**Keywords:** glaucoma; ab interno; minimally invasive glaucoma surgery; cataract surgery; high-frequency deep sclerotomy; intraocular pressure

## 1. Introduction

It is well known that glaucoma comes first in cases of irreversible vision loss and is the second leading cause of blindness worldwide [1]. Nowadays in developed countries, less than 50% of people are unaware of their diagnosis, mainly because of the asymptomatic nature of chronic glaucoma [2].

Trabeculectomy remains the gold standard for glaucoma surgery despite high rates of complications [3–5]. Choroidal effusions, hypotony, shallow anterior chambers, and hyphema are known as early postoperative complications [6–8]. Late complications are often bleb-related, including leakage, blebitis, and endophthalmitis. These complications are more common if antimetabolites like 5-Fluorouracil and Mitomycin C are used [9].

Outflow resistance is caused primarily by the juxtacanilicular trabecular meshwork and, especially, by the inner wall of the Schlemm's canal. This fact is used for the concept of the trabecular meshwork bypass [10]. It is presumed that 35% of the outflow resistance arises in the inner wall of the Schlemm's canal [11].

In addition to the procedures already mentioned, we focus in this work on an enhanced Schlemms canal (SC), which can be addressed both internally and externally. According to the current classification, the ab interno methods include trabectome, iStent, Hydrus, the results presented in this work, and high-frequency deep sclerotomy (HFDS) [12]. The trabectome was introduced in 2004 and has an electroablation of an arc of trabecular meshwork where direct access to the collector channels is given. The intraocular pressure (IOP) reduction of up to 40% is stated in the literature for this ab interno method [13,14]. Another approach for a micro-invasive glaucoma surgery (MIGS) from interno is the iStent. With this device, the aqueous humor is drained directly into the Schlemm's canal, thus avoiding the resistance of the trabecular meshwork [15]. Another representative of the ab interno procedure of the Schemm's canal microstent is the so-called hydrus. It is an 8 mm long crescent-shaped open structure that the bend and sludge canal adapts. It is introduced into the Schlemm's canal by a clear corneal incision [16]. The HFDS is an internally operative glaucoma method that was introduced in 1999. Initially, this surgical procedure was known as sclerothalamotomy ab interno (STT ab interno), and its name was only changed to HFDS in 2012. The aim of the new nomenclature was to describe the procedure more precisely. Regarding the procedure, there were no differences between STT ab interno and HFDS [17–20].

The aim of all MIGS is that, in addition to being effective, they can be carried out simply, quickly and inexpensively with the longest possible effect. The aim of this study is to show the clinical 4-year investigations of this trans-trabecular surgery using the Schlemm's canal as an outflow pathway combined with cataract surgery.

## 2. Materials and Methods

In this single center study from December 2012 till December 2016, 223 patients with insuffiently controlled primary open-angle glaucoma under maximal tolerated topical therapy without history of prior ocular surgery with significant cataract were included. Exlcusion criteria were monophthalmia, angle closure with or without glaucoma, missing willingness to attend follow-up examinations with randomization or any psychiatric disorder, and any condition that affects the optical system (severe alteration of cornea, anterior chamber, or retina). During the observation period of 48 months, 18 patients did not finish the follow up due to several reasons, so finally 205 patients were included in the study. HFDS was performed combined with cataract surgery, with cataract surgery first being performed. Both procedures were always performed by the same surgeon. A complete ophthalmological examination was carried out in each patient prior to surgery. Further ophthalmologic follow-up examinations were carried out postoperatively on days 1, 2, and 3, after 1 and 4 weeks, and then every 3 months until 48 months. Gonioscopy with a three-mirror goniolens was performed after 4 weeks to check the persistence of the sclerotomy. Tobramycin/dexamethasone and pilocarpin 2% eye drops were applied 3 times daily for 1 month. The tenets of the Declaration for Helsinki were followed in this study. The cantonal ethics committee of Aargau approved the study. According to the ethics committee, patient consent is not required for this retrospective study.

*High-Frequency Diathermic Probe*

The high-frequency diathermic probe (abee® Glaucoma Tip, Oertli Instrumente AG, Berneck, Switzerland) consists of an inner and outer coaxial electrode. Both electrodes are isolated, while the inner one is made by platinum.

The dimensions of platinum probe tips are 1 mm in length, 0.3 mm high, and 0.6 mm wide. The platinum probe tip is bent posteriorly at an angle of 150°. The probe external diameter is 0.9 mm. Electrical current is modulated at MF (~500 kHz) frequency. The temperature of approximately 130 °C

at the tip of the probe is generated by alternating current, while target tissue temperature ranges from 90° to 100°. The tissue is heated by radio frequency-induced intracellular oscillation of ionized molecules, which leads to elevation of intracellular temperature. The very high temperatures cause breakdown of tissue molecules. Inhomogeneous electric field with high voltage and selective current flow conduct to electric arcs (Figure 1).

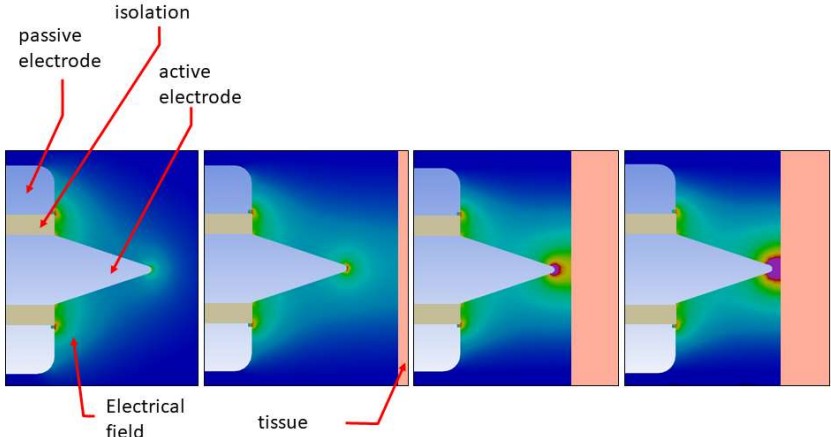

**Figure 1.** Inhomogeneous electric field, high voltage, and electric arc.

Bearing in mind that the vaporization or cutting process is the best accomplished with relatively low voltage, the coagulation is performed by arcing modulated high-voltage current to tissue. The electric arcs create cell bursts through the evaporation of cell content. By modulating the voltage, it is possible to cut and at the same time make locale coagulation. There is a galvanic separation between the power source and the surgery hand piece (Figure 2).

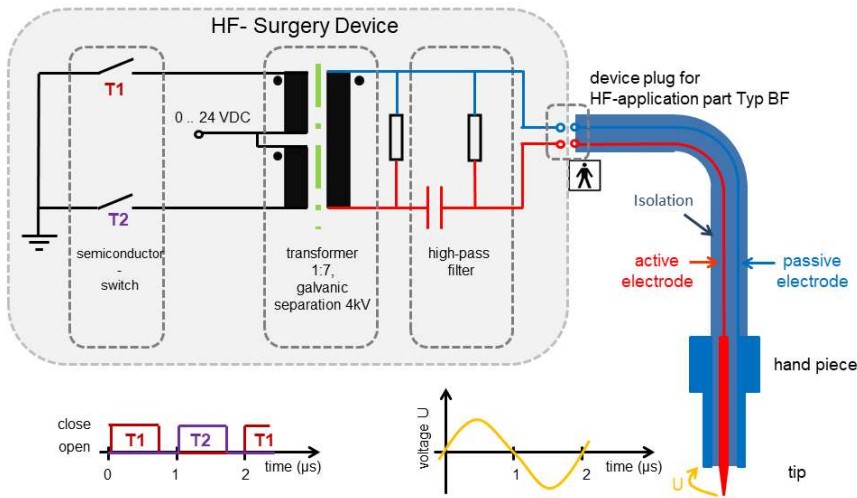

**Figure 2.** High-frequency (HF) generator.

First, the electrodes are placed away from the tissue, and then the gap between two electrodes is ionized. Due to ionization process, an electric arc discharge develops. In this approach, tissue burning is more superficial because the current is spread over the tissue area more than over the tip of electrode. The experimental set-up provides high-frequency power dissipation in the vicinity of the tip. This was demonstrated in the histological analysis by showing that there is more of a cutting effect in the tissue than coagulation, because there is no tissue nor cell denaturation in the cutting channel and surrounding tissue (Figure 3).

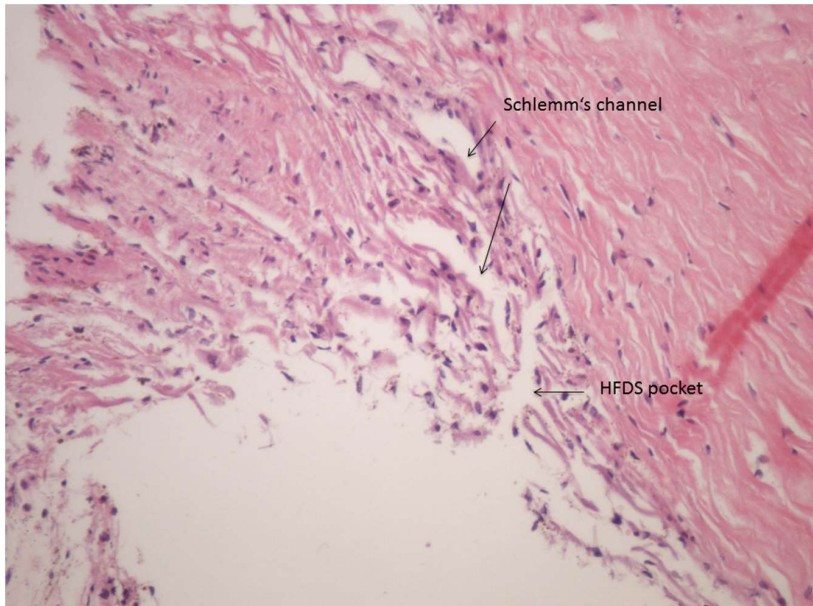

**Figure 3.** Histological analysis of a high-frequency deep sclerotomy (HFDS) cut (HE staining, human eye).

Two clear corneal incisions were used: one of 1.2-mm temporale or temporo-superior for introducing the abee tip and the other nasal with 0.8-mm created for cataract surgery. In the study, miochol (Acetylcholine chlorid 20 mg) was given intracamerally for the miosis. Anterior chamber was filled with a cohesive ophthalmic viscosurgical device (OVD). The standard high-frequency probe (Figure 4) as described above is consecutively inserted through the temporal paracentesis using a four mirror gonioscopic lens, until the probe is properly placed opposite nasally to the iridocorneal structure.

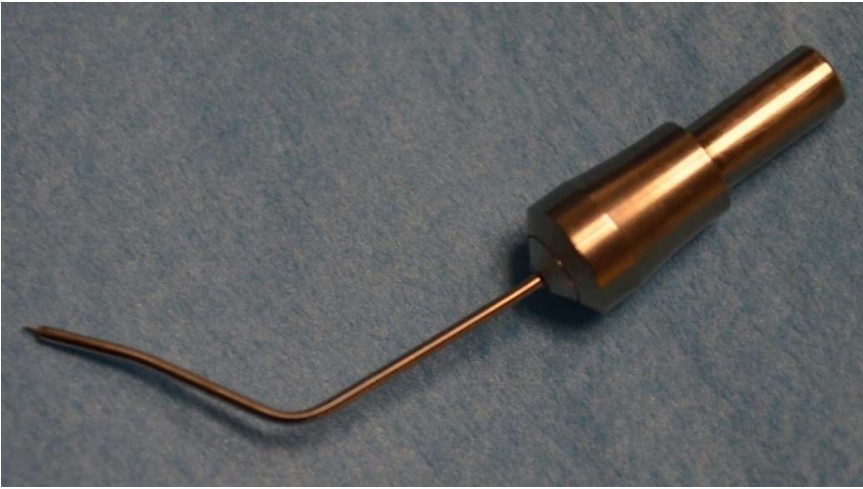

**Figure 4.** Abee® (HFDS) tip.

Then, six pockets were created consecutively in a row with approximately one tip length space between them. Recently, a new diathermic probe design was developed and used for all patients in the study. For all patients, six pockets were done. With the new abee tip design, three accesses to the anterior chamber are possible, i.e., temporal, superotemporal, or superonasal. The pockets can thus be placed nasally, nasally inferior, or nasally temporal. Consequent retreatments can be done easily. The external diameter is 0.9 mm. Oertli devices were used for the study, i.e., the Pharos and Catarhex 3, with the same setting.

The electric specifications of the probe remained unchanged. Pockets were created with the probe 1 mm through the trabecular meshwork, into Schlemm canal. The target was the insertion of a pocket of 1 mm into the sclera. The target deep sclerotomy pocket is approximatively 0.3 mm thick and 0.6 mm wide, resulting in a resorption surface area of 3.6 mm$^2$ (Figure 5a).

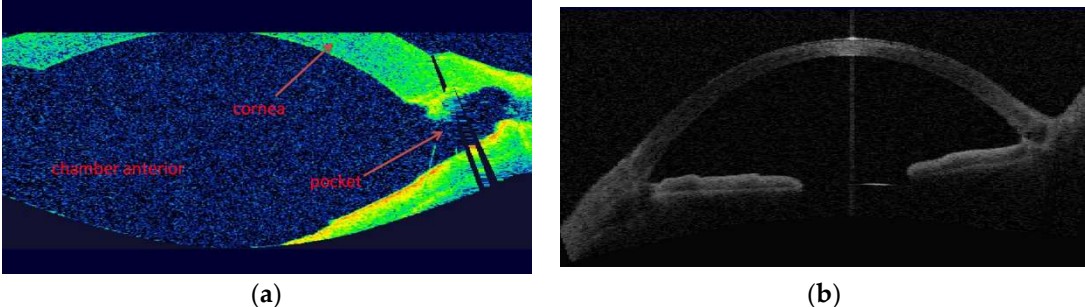

| (**a**) | (**b**) |

**Figure 5.** (**a**) OCT image of the anterior chamber: Overview after HFDS procedure with a newly created pocket in the chamber angle (Visante OCT, Zeiss, Oberkochen, Germany). (**b**) OCT image of the anterior chamber of a normal angle anterior before HFDS surgery.

For comparison, it is shown a normal anterior chamber and angle anterior (Figure 5b).

The tip's dimensions and ab interno approach make it compatible with the stipulations of minimally invasive glaucoma surgery.

Statistical analysis: statistical calculation was done with SPSS Program Version 22. Two-tailed student t-test was used for statistical evaluation of parametric data. Significance was set at a *p* value of <0.05.

## 3. Results

Mean age of patients with open-angle glaucoma was 76.8 ± 11.1 years (range: 35–88 years). 103 patients (50.2%) were female, and 102 patients (49.8%) male. One eye of each patient was included. During the whole period of 48 months, no repeat-surgery was needed in any included patient.

Mean preoperative IOP in the study population of 205 patients was 24.5 ± 4.3 mmHg (range 18 to 48 mmHg). Decimalised Snellen visual acuity (VA) increased from 0.46 ± 0.27 preoperative to 0.68 ± 0.27 postoperative. For all patients, the follow-up was 48 months. After 10 days, a slight increase of IOP was detected but was not statistically significant. Mean IOP after 48 months was 15.0 ± 1.7 mmHg (range 10 to 20 mmHg). The IOP drop was statistically significant ($p < 0.001$) at all measured postoperative intervals (Figure 6).

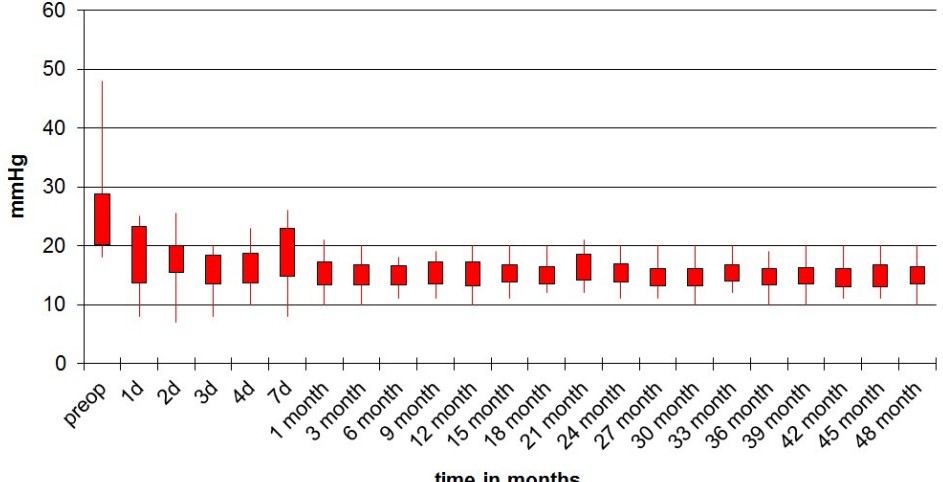

**Figure 6.** Intraocular pressure (IOP) follow-up of operated eyes during 48 months.

At month 48 after surgery, 54.7% of patients had an IOP < 15 mmHg, 77% had an IOP < 18 mmHg, and 84.9% had an IOP < 21 mmHg. Qualified success rate, defined as an IOP lower than 22 mmHg with medication, was 100% for all patients at 48 months (Figure 7).

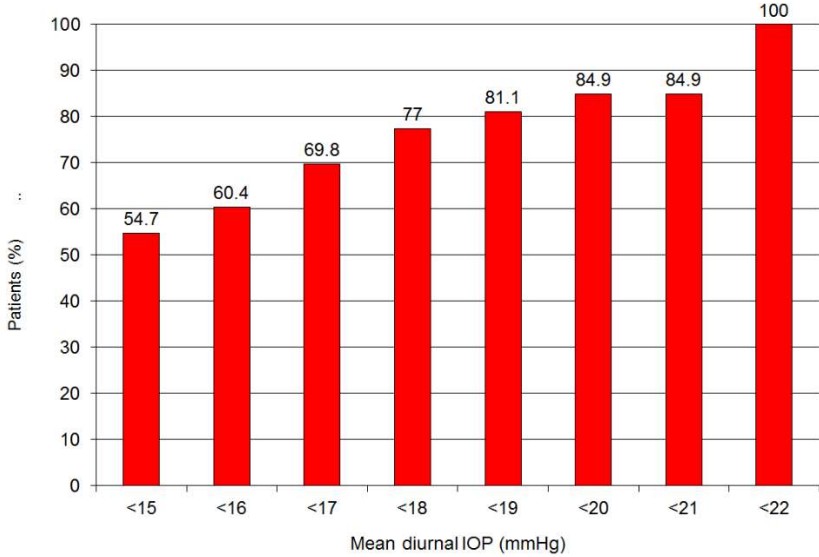

**Figure 7.** Qualified success rate after 48 months of follow-up.

The average preoperative administration of pressure-reducing number of antihypertensive eye agents was 2.6 ± 1.0. Following surgery, this value was decreased to 0.47 ± 0.59 after 1 month, 0.40 ± 0.54 after 3 months, 0.28 ± 0.62 after 6 months, 0.31 ± 0.49 after 12 months, 0.38 ± 0.57 after 24 months, 0.42 ± 0.83 after 36 months, and 0.48 ± 0.97 after 48 months (Figure 8).

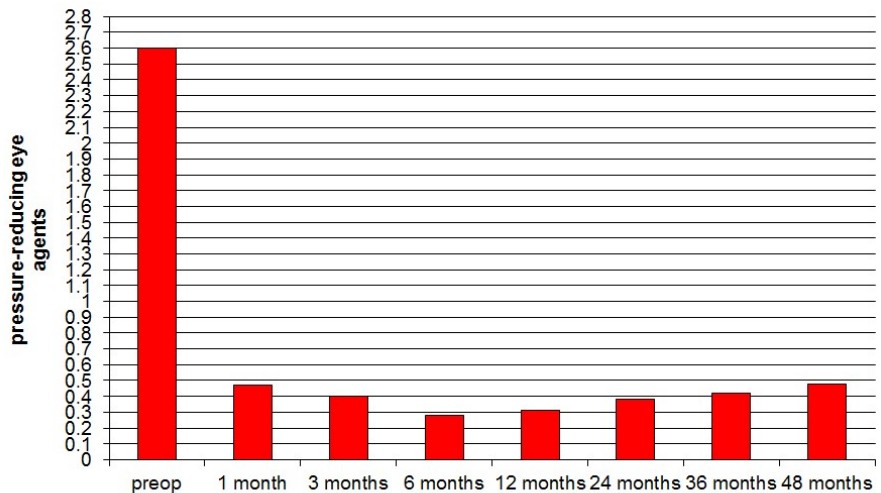

**Figure 8.** Progression of pressure-reducing eye agents pre-and postoperative during the 48 months, showing a marked and sustained postoperative reduction in numbers of agents needed.

There were no significant visual field changes (Octopus 900, Haag Streit, Switzerland) from baseline with a mean defect MD 7.06 ± 6.54 and loss of variance LV 26.4 ± 23.92. At 48 months, these values were MD 8.43 ± 2.11 and LV 24.1 ± 24.92 ($p = 0.22$ for MD, $p = 0.58$ for LV).

The mean excavation preoperative was 0.63 ± 0.22 and after 48 months 0.65 ± 0.21, which is not statistically significant ($p = 0.36$).

Temporary IOP elevation higher than 21 mmHg was observed in 18 of 205 eyes (8.7%). Four eyes (2%) showed transient fibrin formation. Fibrin was treated by topical dexamethasone and disappeared one day later.

## 4. Discussion

The present study has shown sustained IOP—lowering effect over 48 months using Schlemm's canal as an outflow pathway with a novel MIGS procedure. Looking into the past, the only possibility of glaucoma treatment was conservative medical management and more invasive glaucoma surgery. The advance MIGS procedure was intended to fill the existing gap regarding treatment. Early studies [19] have demonstrated its ability to lower IOP with minimal risk for mild to moderate glaucoma.

The HFDS ab interno method intends the creation of a direct channel between the anterior chamber and the Schlemm canal. Persistence of the sclerotomy has been investigated with a three-mirror goniolens 4 weeks after the procedures. The abee tip creates a deep sclerotomy with subsequent access of aqueous outflow to the scleral layer. Both aspects may facilitate a bypass effect of aqueous outflow. In an earlier study with deep sclerotomy ab interno, a significant IOP peak could be seen ten days postoperatively [17]. As shown in our study, with introduction of postoperative pilocarpine 2% eye drops application for the first 4 weeks, the high IOP peak amplitude could be avoided. This may have also been the case because the procedures were done combined with cataract surgery.

In general, MIGS procedures share five important features: ab interno approach, potential minimal trauma, ability to lower IOP, a high level of safety, and faster visual recovery. Hence, advantages of HFDS ab interno method, compared with trabeculectomy and perforating and nonperforating deep sclerectomy, seem to have a low rate of postoperative complications and a stable level of reduced IOP. Hypotension, a frequent finding in trabeculectomy, which is less frequently found in nonperforating deep sclerectomy, was not seen in the present population. The most frequent early complications in trabeculectomy are hyphaema (24.6%), shallow anterior chamber (23.9%), hypotony (24.3%), wound leak (17.8%), and choroidal detachment (14.1%). The most frequent late complications are iris incarceration (5.1%) and encapsulated bleb (3.4%) [21]. After HFDS, a transient IOP elevation was seen in 18 of 205 eyes (8.7%), on average occurring 10 days postoperatively. Four of 205 eyes (2%) had transient fibrin formation. Therefore, compared with penetrating or nonpenetrating techniques, HFDS seems to be a safe surgical technique [22–25].

An earlier study with deep sclerotomy ab interno had a complete success rate of 83% after 48 months [17,19] for open angle glaucoma. The present study shows that the success rate of 84.9% was similar. The slight improvement could be due to the fact that the procedures was carried out in combination with cataract surgery. It was particularly striking that the postoperative IOP range was significantly lower in the combined procedure. This effect could be explained by the additional reduction due to the cataract surgery, but also by the postoperative application of pilocarpine 2% during the first 4 weeks. If the postoperative IOP profile is analyzed, after 24 months there is a loss of the additional IOP-lowering effect from the cataract surgery will be apparent.

Advantages of HFDS include its comparative simplicity and short duration of the surgical procedure itself. Additionally, unlike ab externo filtering techniques, the technique avoids stimulation of episcleral and conjunctival tissues leading to fibroblast activation. Additionally, by comparing the histological alterations of HFDS with trabectome, another MIGS procedure, it has been shown that with trabectome tissue was damaged near the incision [26]. As trabectome was validated mainly in the pediatric population associated with long-term IOP control, recently a dual-blade device was developed for adult use but showing also smaller but still detectable damage to the tissue [27].

## 5. Conclusions

Although the number of six surgical pockets chosen was arbitrary, our results suggest that it may be sufficient to provide good long-term IOP-lowering efficacy and safety. With the introduction of the new abee-tip design, it is possible to perform surgical retreatment in the nasal and inferior quadrants

of the trabecular meshwork. HFDS is a safe and minimally invasive method for glaucoma surgery with good long-term results. More studies are needed to confirm our findings in different populations and types of glaucoma.

**Author Contributions:** B.P. provided the treatment indication, developed the study design, acquired clinical data, and contributed to writing the paper. Z.C. contributed substantially to the development of the study design and substantially contributed to writing the paper. K.M. performed data analyses and substantially contributed advising regarding the paper contents. M.R. performed data anlysis and substantially contributed the study design. R.A. performed data analysis and contributed to writing the paper. All authors have read and agreed to the published version of the manuscript.

**Funding:** This research received no external funding.

**Conflicts of Interest:** The authors declare no conflict of interest.

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
