# Peer review of "High-Frequency Deep Sclerotomy, A Minimal Invasive Ab Interno Glaucoma Procedure Combined with Cataract Surgery: Physical Properties and Clinical Outcome"

_applsci, doi:10.3390/app10010218_

Round 1
Reviewer 1 Report
The purpose and goal of the present study has not been properly identified. What new information the present research is going to decipher. Abee HDFS probe’s intended use is MIGS and can be used in combination with cataract surgery. The present manuscript reports the outcome of both glaucoma surgery in combination with the cataract surgery in a moderately large patient cohort (205). The authors for the present manuscript are pioneer in this specific research area and they have consistently published updated cutting-edge MIGS procedures employing Abee HDFS probe and CataRhex 3 equipment. The present research convincingly proved a consistent lowering of IOP compared with the pre-surgical values.
Major:
One specific set of data is missing in the present manuscript where glaucoma surgery is done with the probe but no cataract surgery has been performed. Inclusion of this data will add certainly more value to the paper in showing the combined surgical procedure could be more effective in lowering IOP than the glaucoma surgery alone.From the present manuscript, it was not quite clear whether CataRhex 3 has been used to perform the combined surgery.
Page 1, lines 19-21: Background information of the abstract is not clear. What are the authors trying to say?
No information for the approved IRB protocol or the signed informed consent have been found elsewhere in the manuscript.
Fig 4: The image of Abee HDFS probe is not a data figure. This probe is commercially available. Referring the probe is sufficient. If the authors want to include the image, please include as supplementary figure.
Minor:
Page 1, line 38: delete ‘in the’ Page 1, line 41: Change ‘golden standard’ to ‘gold standard’ Page 2, line 46: Change ‘juxtacanicular’ to ‘juxtacanilicular’ Page 2, lines 50-54: ‘The present procedure we described already in a formerly publication [12] … pathway for aqueous humor the subconjunctival space, (Xen Gel Stent, Allergan). This section does not make any sense. This is a bery poorly written paragraph. Please revise and rewrite. Page 3, line 94: Correct ‘…electrodes is ionized.’ to ‘…electrodes are ionized.’ Page 4, line 103: ‘0.8 mm’ what has been created? Page 4, line 103: Correct ‘chamber anterior’ to ‘anterior chamber’. Page 4, lines 110-112: ‘With the new design the 110 surgery…superotemporal, 111 or superonasal.’ This statement does not make any sense. Please revise and rewrite. Page 6, line 152-153: ‘4 eyes (2%) shows transient fibrin formation.’ Correct the statement to ‘4 eyes (2%) show transient fibrin formation.’ Page 6, line 142: What is ’anti-hypertensive eye agent’? Page 7, line 158: Change ‘medical management’ to ‘clinical management’. Page 7, line 159: Correct ‘gab’ to ‘gap’. Explain all the abbreviations used when they appear first. Like ‘micro-invasive glaucoma surgery (MIGS)’.Author Response
Thank you very much for the very valuable comments of the reviewer. The work has gotten so much better.
The purpose and goal of the present study has not been properly identified. What new information the present research is going to decipher. Abee HDFS probe’s intended use is MIGS and can be used in combination with cataract surgery. The present manuscript reports the outcome of both glaucoma surgery in combination with the cataract surgery in a moderately large patient cohort (205). The authors for the present manuscript are pioneer in this specific research area and they have consistently published updated cutting-edge MIGS procedures employing Abee HDFS probe and CataRhex 3 equipment. The present research convincingly proved a consistent lowering of IOP compared with the pre-surgical values.
In the abstract and in the work itself, we specified the goal he present study. Thank you very much for the remark.
We wanted to show the effectiveness of HFDS using a larger patient population, also in combination with cataract surgery. We used two Oertli devices for the study, the Pharos and CataRhex 3, however, with the same setting, both in terms of HFDS and cataract surgery. We added this addition in the paper.
Major:
One specific set of data is missing in the present manuscript where glaucoma surgery is done with the probe but no cataract surgery has been performed. Inclusion of this data will add certainly more value to the paper in showing the combined surgical procedure could be more effective in lowering IOP than the glaucoma surgery alone.
In the following publication, we performed an exclusive HFDS without cataract surgery in 53 patients with a follow-up of 72 months:
Pajic B, Pajic-Eggspuehler B, Haefliger I. New minimally invasive, deep sclerotomy ab interno surgical procedure for glaucoma, six years of follow-up. J Glaucoma 2011 Feb;20(2):109-14.
In the same way as in the current work, all surgeries in this cited paper were performed by the same surgeon. In this sense, the results can be compared with each other to discuss the effect of the combined HFDS cataract surgery. The mean IOP was slightly lower in the combined HFDS / cataract surgery than in the HFDS itself. It was particularly striking that the postoperative IOP range was significantly lower in the combined procedure. This effect could be explained by the additional reduction due to the cataract surgery, but also by the consistent postoperative application of Pilocarpine 2% during the first 4 weeks. If you look at the postoperative IOP profile, after 24 months you lose the additional IOP-lowering effect from the cataract surgery. This part was included in the discussion in the current paper. Thanks very much for the input.
From the present manuscript, it was not quite clear whether CataRhex 3 has been used to perform the combined surgery.
Thank you for this remark. We have used the Pharos and CataRhex 3 device for the study with the identical settings. We have added in the paper.
Page 1, lines 19-21: Background information of the abstract is not clear. What are the authors trying to say?
With all the changes and reformulations in the introductions, we had the opportunity to clarify the content and rewrite it precisely (see corrected script). Many thanks for the valuable input.
No information for the approved IRB protocol or the signed informed consent have been found elsewhere in the manuscript.
Unfortunately, this was forgotten in the first paper version. Of course, we listed it in the revision version (see corrected script). Many thanks.
Fig 4: The image of Abee HDFS probe is not a data figure. This probe is commercially available. Referring the probe is sufficient. If the authors want to include the image, please include as supplementary figure.
A new image from Abee HDFS has been added. The new authentic image has replaced Figure 4. Thank you for this input.
Minor:
Page 1, line 38: delete ‘in the’ Page 1, line 41: Change ‘golden standard’ to ‘gold standard’ Page 2, line 46: Change ‘juxtacanicular’ to ‘juxtacanilicular’ Page 2, lines 50-54: ‘The present procedure we described already in a formerly publication [12] … pathway for aqueous humor the subconjunctival space, (Xen Gel Stent, Allergan). This section does not make any sense. This is a bery poorly written paragraph. Please revise and rewrite. Page 3, line 94: Correct ‘…electrodes is ionized.’ to ‘…electrodes are ionized.’ Page 4, line 103: ‘0.8 mm’ what has been created? Page 4, line 103: Correct ‘chamber anterior’ to ‘anterior chamber’. Page 4, lines 110-112: ‘With the new design the 110 surgery…superotemporal, 111 or superonasal.’ This statement does not make any sense. Please revise and rewrite. Page 6, line 152-153: ‘4 eyes (2%) shows transient fibrin formation.’ Correct the statement to ‘4 eyes (2%) show transient fibrin formation.’ Page 6, line 142: What is ’anti-hypertensive eye agent’? Page 7, line 158: Change ‘medical management’ to ‘clinical management’. Page 7, line 159: Correct ‘gab’ to ‘gap’. Explain all the abbreviations used when they appear first. Like ‘micro-invasive glaucoma surgery (MIGS)’.
All corrections were systematically improved in the script. They are marked in the text. Many thanks for the help.
Reviewer 2 Report
The study is interesting and contains the method of deep sclerotomy, a procedure which has already been presented in other publications.
- 1.Introduction line 49: A brief presentation of the literature on the MIGS used should be included, for example as swown in the article:
The Efficacy and Safety of High-Frequency Deep Sclerotomy in Treatment of Chronic Open-Angle Glaucoma Patients.
BioMed Research International Volume 2019, Article ID 1850141, 7 pages
https://doi.org/10.1155/2019/1850141
- 2. Materials and Methods
line 57: the period of recruitment of patients included in the study should be indicated
line 104:since the sclerotomy was performed after cataract surgery (as written in the abstract) it must be specified if it was induced myosis pharmacologically before treatment
line 110: "A new diathermic probe design wae developed".It is not clear whether the diathermic probe was modified during the study and whether this led to a change in the results obtained. Moreover, 6 or 3 surgical pockets were done?
line 120: In addition to the OCT image in Fig.5, an other image (photo) of the trabecular meshwork before and after treatment is useful.
- 3. Results
line 138: From Fig.7 we can see that 100% of the patients have an IOP lower than 22 mmHg. Did all patients have a significant reduction in IOP?
line 149: "..non significant visual field changes.." In methods it's not specify the staging of the recruited glaucomas
Author Response
The study is interesting and contains the method of deep sclerotomy, a procedure which has already been presented in other publications.
- 1.Introduction line 49: A brief presentation of the literature on the MIGS used should be included, for example as swown in the article:
The Efficacy and Safety of High-Frequency Deep Sclerotomy in Treatment of Chronic Open-Angle Glaucoma Patients.
Mokhtar Mohamed Ibrahim Abushanab, Ayman El-Shiaty, Tarek El-Beltagi,and Shaymaa Hassan Salah
BioMed Research International Volume 2019, Article ID 1850141, 7 pages
https://doi.org/10.1155/2019/1850141
Thank you for the valuable information. The introduction was rewritten and the recommended paper cited.
- 2. Materials and Methods
line 57: the period of recruitment of patients included in the study should be indicated
Done
line 104:since the sclerotomy was performed after cataract surgery (as written in the abstract) it must be specified if it was induced myosis pharmacologically before treatment
done and was added at work
line 110: "A new diathermic probe design wae developed".It is not clear whether the diathermic probe was modified during the study and whether this led to a change in the results obtained. Moreover, 6 or 3 surgical pockets were done?
This is an important remark! Thanks. There were 6 pockets that were performed. This information was introduced accordingly in the work.
line 120: In addition to the OCT image in Fig.5, an other image (photo) of the trabecular meshwork before and after treatment is useful.
As figure 5b an OCT image was inserted with the state before an HFDS
- 3. Results
line 138: From Fig.7 we can see that 100% of the patients have an IOP lower than 22 mmHg. Did all patients have a significant reduction in IOP?
I have to clarify on this question that this is a Qualified success rate, defined as an IOP lower than 22 mmHg with medication. Initially, the IOP decreased significantly in all eyes, but after a certain period one had to start again in some cases with medication that after 48 months was 0.48 ± 0.97 pressure-reducing number of anti-hypertensive eye agents. Your input is very valuable. Thanks a lot!
line 149: "..non significant visual field changes.." In methods it's not specify the staging of the recruited glaucomas
That is an excellent remark. Due to the middle defect (MD) from the visual field we already have a grading of the glaucoma. However, it is important to consider the excavation of the papilla at this point. The excavation preoperative and 48 months postoperative has now been listed in the paper. This fits perfectly to the visual fields, where no statistical change could be found in the course. Thank you very much for this input.
Round 2
Reviewer 1 Report
I am satisfied with the responses provided by the authors. They have taken an adequate effort in order to revise the manuscript following my suggestions and advice. The manuscript has been improved significantly.
Reviewer 2 Report
The revised text was well done and this has greatly improved the understanding of the work.
Referring to figure 5 a and b, a photo of the trabecular meshwork before and after the sclerotomy would have been useful, but also the OCT explains the meaning of the treatment quite well.